# Patterns of Genital Tract *Mustelid Gammaherpesvirus 1* (Musghv-1) Reactivation Are Linked to Stressors in European Badgers (*Meles Meles*)

**DOI:** 10.3390/biom11050716

**Published:** 2021-05-11

**Authors:** Ming-shan Tsai, Sarah François, Chris Newman, David W. Macdonald, Christina D. Buesching

**Affiliations:** 1Recanati-Kaplan Centre, Wildlife Conservation Research Unit, Department of Zoology, University of Oxford, Abingdon Road, Tubney House, Tubney, Oxfordshire OX13 5QL, UK; chris.newman@lmh.ox.ac.uk (C.N.); david.macdonald@zoo.ox.ac.uk (D.W.M.); 2Evolve.Zoo, Peter Medawar Building for Pathogen Research, Department of Zoology, University of Oxford, South Park Road, Oxford OX1 3SY, UK; sarah.francois@zoo.ox.ac.uk; 3Cook’s Lake Farming Forestry and Wildlife Inc. (Ecological Consultancy), Queens County, NS B0J 2H0, Canada; christina.buesching@ubc.ca; 4Department of Biology, Irving K. Barber Faculty of Science, University of British Columbia, Kelowna, BC V1V 1V7, Canada

**Keywords:** sexually transmittable infections, wildlife disease, epidemiology, seasonal effect, weather conditions

## Abstract

Gammaherpesvirus reactivation can promote diseases or impair reproduction. Understanding reactivation patterns and associated risks of different stressors is therefore important. Nevertheless, outside the laboratory or captive environment, studies on the effects of stress on gammaherpesvirus reactivation in wild mammals are lacking. Here we used *Mustelid gammaherpesvirus 1* (MusGHV-1) infection in European badgers (*Meles meles*) as a host–pathogen wildlife model to study the effects of a variety of demographic, physiological and environmental stressors on virus shedding in the genital tract. We collected 251 genital swabs from 150 free-ranging individuals across three seasons and screened them for the presence of MusGHV-1 DNA using PCR targeting the DNA polymerase gene. We explored possible links between MusGHV-1 DNA presence and seven variables reflecting stressors, using logistic regression analysis. The results reveal different sets of risk factors between juveniles and adults, likely reflecting primary infection and reactivation. In adults, virus shedding was more likely in badgers in poorer body condition and younger than 5 years or older than 7; while in juveniles, virus shedding is more likely in females and individuals in better body condition. However, living in social groups with more cubs was a risk factor for all badgers. We discuss possible explanations for these risk factors and their links to stress in badgers.

## 1. Introduction

Alongside the *Alphaherpesvirinae, Betaherpesvirinae* and *Deltaherpesvirinae*, the Gammaherpesvirnae is a subfamily of double-stranded DNA viruses belonging to the family Herpesviridae [1,2] containing four genera, the *Macavirus*, *Rhadinovirus*, *Lymphocryptovirus* and *Percavirus*. After primary infection, most known gammaherpesviruses establish a latent stage in the B lymphocytes, and can then be reactivated repeatedly throughout life. Numerous factors are known to trigger reactivation, including stress [3,4], trauma (e.g., surgery: [5]) or primary co-infection with other pathogens [6]. Reactivation is a process of viral lytic infection, which involves virus replication within the host cell, eventually causing the cells to burst and release infectious virions. Reactivation of gammaherpesviruses occurs in plasma cells and epithelial cells of mucosa that function as portals for external contact (e.g., mouth, nose, eyes and genital tract), thus facilitating transmission [7]. Reactivation is typically asymptomatic or induces only mild disease, but, depending on strain pathogenicity [8], can also contribute to the development of other severe diseases such as cancer [9], and is associated with a higher risk of contracting co-infections with additional highly virulent pathogens (e.g., herpesvirus DNA detection is a risk factor for the presence of *Chlamydia pecorum* DNA in koalas [10]) or pathogens causing immunodeficiency (e.g., Human herpesvirus 8 and HIV [11]). Although not as extensively studied as the human Epstein-Barr Virus (EBV) and Kaposi’s sarcoma-associated herpesvirus (KSHV), gammaherpesvirus reactivation has long been linked to sometimes severe diseases in domestic animals [12,13,14], and also increasingly to illness in wildlife species [15,16,17,18].

European badgers, *Meles meles* (henceforth ‘badgers’), are seasonally breeding mustelids that are commonly infected with the *Mustelid gammaherpesvirus 1* (MusGHV-1, genus *Percavirus*), for which prevalence of viral DNA in blood samples can reach up to 100% in the UK and in Ireland across all age groups, with primary infection likely occurring in early life [19,20]. Furthermore, MusGHV-1 DNA has also been detected in 55–82.5% badger genital swab samples [21,22], suggesting local genital tract reactivation is common, and thus MusGHV-1 is a sexually transmittable pathogen. Sexually transmitted infections (STIs) are often associated with infertility in animals [23,24], and in badgers previous research has linked otherwise asymptomatic MusGHV-1 shedding in genital tracts to poorer female reproductive performance [22]. Considering the potentially considerable impact of STI-induced sterility on population dynamics and potential to drive extinction [25,26], understanding the patterns of MusGHV-1 genital tract reactivation is important in identifying the risk factors. Two previous studies on genital tract MusGHV-1 screening have indicated that cubs, old individuals and adult males during the mating season are at particular risk of genital MusGHV-1 shedding [21,22]. Nevertheless, the impact of social, physiological and seasonal effects, which may contribute to immune dysregulation due to stress, on MusGHV-1 reactivation has not been investigated.

Since the local extinction of brown bears (*Ursus arctos*) and wolves (*Canis lupus*) [27], badgers do not have natural predators in the UK, and their survival is mainly dependent on weather conditions [28,29], which are closely linked to food availability. Faecal corticoid levels indicate that badgers experience seasonal variation in stress levels [30,31]. There are several potential underlying causes: seasonal stress may be linked to seasonally varying food availability [30,31] that can result in mortality [28,29]. Higher cortisol levels are also associated with poor body condition and coinfection with *Mycobacterium bovis*, the pathogen causing bovine tuberculosis [32]. Therefore, metrics of body condition, especially reduced body-condition as a consequence of recent lactation [30,33], can indicate that individuals experience physiological stress [30]. Sociologically, higher social group density is associated with female reproductive suppression [34,35], reduced body condition and fecundity [36] and increased bite wounding among male badgers [37]. Furthermore, aging reduces tolerance to stress [38], specifically altering the balance between innate and acquired immunity in badgers [39], and increasing their risk of herpesvirus reactivation [22], as also observed in other carnivora species [40,41], sometimes resulting in chronic or continuous herpesvirus reactivation [42].

Here we conducted population-wide molecular screening using genital swabs taken from a free-ranging badger population in the south of England across three seasons (spring: May, summer: September, autumn: November). We investigated whether environmental factors (i.e., season, social group size and structure), host demographic parameters (i.e., sex, age and lactation) and host body conditions are risk factors for genital MusGHV-1 shedding. Furthermore, we investigated if risk factors differ between juveniles and adults, given that virus shedding in juveniles is more likely associated with primary infection, whereas in adults it is likely due to reactivation from latency.

## 2. Materials and Methods

Samples were collected from live-trapped badgers in Wytham Woods, Oxfordshire, UK (51°46′26″ N, 1°19′19″ W; caught in May, September and November 2018 following the methodology described in Macdonald et al. [43]; for details see Table 1). For each capture, we recorded sex, sett (i.e., communal den used by a social group) of capture, body condition score (BCS, categorised as 1 = very thin to 5 = very fat) and lactational status (determined by teat measurements of females in spring: [44]). Because each badger in Wytham is given an individual tattoo at first capture (usually as a cub [45]), exact age (in years) was known for most (243 of 251) animals in the dataset. For the remaining eight badgers first caught as adults, age was inferred by toothwear according to the method described by Bright Ross et al. [46]. We defined four age classes based on sex-steroid levels [47] and age: (i) juveniles < 2 years old (cubs and yearlings were combined to increase sample sizes, as there was no difference between MusGHV-1 prevalence in cubs and yearlings: Fisher’s exact test: *p*-value = 0.7449); (ii) young adults: 2 ≤ *x* < 5 years old; (iii) old adults: 5 ≤ *x* < 8 years old; (iv) very old adults: ≥8 years old. The number of cubs and adults resident in each sett was estimated using minimum number alive (MNA) estimates [43,46].

Sterile cotton tops with wooden shafts were used to swab the genital tracts of all females (cubs and adults) and all males (except for very small male cubs in spring for animal welfare reasons), and stored in 2 mL sterile microcentrifuge tubes. All samples were frozen and stored at −20 °C immediately after sampling. Badgers were released at their site of capture on the same day, after full recovery from anaesthesia.

Each swab was reconstituted with 400 µL sterile double distilled water and vortexed gently at room temperature for 10 min. A 200 µL aliquot was taken from the reconstituted swab fluids, and viral DNA was extracted and purified using a commercial kit (DNeasy Blood and Tissue Kit, Qiagen, Hilden, Germany) following manufacturer’s instructions. Purified DNA was then eluted in 100 µL of the provided buffer.

The purified DNA was screened using a MusGHV-1-specific primer pair designed by King et al. 2004 [20], targeting 281 base pairs of the partial DNA polymerase gene. For each reaction, a total of 20 µL PCR solution was mixed with 10 µL HotStartTaq Master Mix (Qiagen, containing 1 unit of HotStartTaq DNA Polymerase, 12 µM of MgCl_2_ and 1.6 µM of each dNTP), 0.5 µM of each primer and 2 µL CoralLoad gel loading dye and 5 µL DNA template. Amplification conditions were kept at 95 °C for 5 min to activate DNA polymerase, followed by 45 cycles of denaturation at 95 °C for 45 s, primer annealing at 60 °C for 45 s, and chain elongation at 72 °C for 1 min, followed by a final extension at 72 °C for 10 min. Finally, the PCR products were loaded in 2% agarose gel to check the amplification results under UV light. Five randomly selected samples with positive results were then amplified again with substituted front primer (5′ CCA AGC AGT GCA TAG GAG GT 3′) to generate longer sequences (771 base pairs). PCR products were then purified and sent for genotyping using Sanger sequencing to confirm the identity of produced amplicon. Sequences returned were then aligned by Clustal W method [48] and analyzed for variation using MEGA X (10.1.7) [49]. Representative sequences were selected and deposited on GenBank under accession numbers MT332100 and MT332101.

Statistical analyses were performed with the R and R Studio software (version 1.21335) [50,51]. Detection rate of genital MusGHV-1 DNA was calculated by dividing the number of PCR positive cases by the total number of tested cases, and 5% upper and lower confidence intervals were calculated using the Wilson method [52]. Logistic regression (glmer function, R package lme4) with badger identity (tattoo) number as a random effect was used to measure univariate effects of MusGHV-1 reactivation in genital tracts with season, sex, age class, BCS, number of residents per social group (total, adult and cub MNA), and percentage of cubs per social group (calculated by cub MNA divided by total MNA) as fixed effect variables. We categorised the percentage of cubs per social group into low and high using 30% as the dividing point according to the distribution of these data (Appendix A). Effects of lactation were analyzed using Fisher’s exact tests, due to low sample sizes. Because juveniles (especially cubs) are generally thinner than adults, and thus have a different body condition distribution [36], in the univariate analysis we analyzed the association of MusGHV-1 detection and individual BCS for juveniles and adults separately. The effect size of each variable is presented using odds ratios. Variables with *p*-values < 0.1 were retained in the multivariable analysis. We constructed two multivariable models based on data from juveniles and adults as we expected patterns to be different between these two groups. Manual backwards selection was used to select the final model with best fit (determined by lowest AIC value). Model residual diagnostics were conducted using the R package DHARMa (version 0.3.3.0). Model fit was established using area-under-receiver-operating characteristics (AUC) with an acceptance threshold of 0.7 [53]. Collinearity of variables was checked using variance inflation factor (VIF), where variables with a VIF value of more than 5 were excluded from the model. We defined variables in the final models with *p*-values of less than 0.1 as risk factors for genital tract MusGHV-1 shedding.

## 3. Results

### 3.1. Univariate Analysis

The overall detection rate of genital tract MusGHV-1 DNA was 35.5% (89/251, 95% CI: 30–41.6%), and was generally higher in summer (43.8%, 35/80) than in spring (34.4%, 33/96) and autumn (28%, 21/75) (Table 2). There was strong evidence for an effect of age on genital MusGHV-1 shedding, where genital tract MusGHV-1 DNA detection rate in juveniles (45.2%, 33/73) and very old badgers (47.7%, 21/44) was higher than in middle-aged adults (i.e., young (30.9%, 30/97) and old (13.5%, 5/37) individuals; Table 2). However, no effect of sex was observed (logistic regression analysis, *p* = 0.679). Adults with lower BCS had a higher probability of genital MusGHV-1 DNA detection rate (*p*-value = 0.012, Table 2). There was no evidence that the number of adults, or the total number of badgers resident in each social group, affected MusGHV-1 DNA detection rates, except for the number of cubs per social group (Table 2): the prevalence of genital tract MusGHV-1 was significantly higher in badgers living in social groups comprising >30% cubs (logistic regression analysis, *p*-value = 0.004). Finally, there was no evidence for recent lactation affecting the MusGHV-1 DNA detection rate in sexually mature females (Fisher’s exact test, *p*-value = 1; Table 2).

### 3.2. Multivariable Analysis

We generated multivariate models of MusGHV-1 detection for juveniles (n = 72) and adults (n = 171). The most parsimonious models for juveniles (Table 3; Appendix A) and adults (Table 4; Appendix A) had both diagnostically acceptable [53] AUC area of 0.727 and 0.774, respectively. In juveniles, multivariate analysis showed that being a female, being in better body condition and living in a social group with a high proportion of cubs (over 30%) were risk factors for genital MusGHV-1 shedding (Table 3). In contrast, in adults, young (2–5 years old) or very old (≥8 years) badgers in poorer body condition, and living in social groups with a high proportion of cubs, are at particular risk of shedding MusGHV-1 in their genital tract (Table 4).

### 3.3. Genetic Diversity of MusGHV-1 in the Wytham Badger Population

We sequenced five randomly selected MusGHV-1 positive PCR products of the partial DNA polymerase gene. All sequences were trimmed to 694 base pairs and confirmed to be MusGHV-1 according to the NCBI BLAST results, returning 98.7% (n = 3) and 100% (n = 2) nucleotide identity to the published MusGHV-1 sequence isolated from a badger in Cornwall, England (accession number: AF275657).

## 4. Discussion

In this study, we investigated the patterns of MusGHV-1 shedding in badger genital tracts, and identified associated risk factors comprising age, sex, season and social group composition. We confirmed that juveniles and adults are subject to different sets of risk factors, putatively reflecting primary infection and reactivation from latency, respectively.

In terms of age class effects, the high genital reactivation rate detected in cubs and yearlings suggests that badgers contract MusGHV-1 early in life, before reaching sexual maturity. Although MusGHV-1 reactivates repeatedly throughout life, reactivation tends to be less frequent in young and old adults, although rates increase in very old individuals. This matches patterns in humans where most people become infected with herpes during their childhood/adolescence (e.g., 100% and 70% seroprevalence of EBV before age 14 in Hong Kong and the United Kingdom [54]), respectively, then typically experience viral latency during their prime, but can suffer from increasingly longer and more frequent herpesvirus reactivation that can cause mild disease (e.g., shingles [55] in old age due to lowered immune response [56,57]).

Since vertical transmission of MusGHV-1 through the placenta is unlikely [22], and the potential for infection from the vaginal tract during parturition is equally low, due to low genital MusGHV-1 detection rate in pregnant females [22], we hypothesise that cubs contract MusGHV-1 through close contact with virus-shedding conspecifics [22]. Thereafter genital virus shedding in cubs may arise after primary acute infection through non-sexual routes and subsequent latency, as observed in the murine model where *Murine herpesvirus 4* (MuHV-4, also a gammaherpesvirus), inoculation in the nasal cavity results in acute infection in the respiratory tract and lungs, and establishes latency in the spleen, but then reactivates in the vaginal tract 17–21 days after inoculation [58]. MuHV-4 nasal cavity inoculation, however, does not result in reactivation in male genital tracts, and transmission is only possible from females to males. After sexual intercourse with virus-shedding female mice, the virus replicates in the male penis for 3 weeks. Interestingly, also in badgers, female juveniles are at higher risk of MusGHV-1 reactivation in the genital tract than males. Furthermore, juveniles in better body condition, regardless of sex, exhibit higher virus detection rate. Indeed, juvenile males in better body condition enter puberty earlier than thinner males (11 months compared to 22–28 months: [59]). Once juveniles enter puberty they will experience an increased risk of contracting MusGHV-1 through sexual contact and/or that their latent infection is reactivated through mating resulting in viral shedding in the genital tract.

Faecal corticosteroid measurements from badgers in Ireland [30] evidence higher stress levels in summer likely associated with dry environmental conditions that result in lower earthworm availability (i.e., the badgers’ main food type [45]); similarly, in our own study population, summer drought is an established mortality factor due to malnutrition/starvation [28,29]. This corresponds to our finding that, in all adults, seasonal MusGHV-1 reactivation rates were highest in summer, but females tended to have higher viral reactivation levels than males in spring—possibly due to reproductive stresses, while the reverse was true in autumn (Table 2) [30]. Badgers captured in summer generally have lower BCS than in other seasons. This is reflected in our multivariate model (Table 4) such that when season and BCS are both included in the model, the effect of seasons found in univariate model (Table 1) is eliminated, while BCS maintains a strong negative effect on MusGHV-1 DNA presence in the genital tract. This implies that BCS is the main driver causing the seasonal effect of MusGHV-1 DNA detection rate in the genital tract. However, we did not detect any collinearity between seasons and BCS, and removing season from the model decreased the model fit (i.e., an increase in the AIC value), suggesting season contributed to the response value through a different explanatory variable that was not included in our study.

Chronic stress has proven a significant risk factor of herpesvirus reactivation, causing immune system dysregulation, where corticosteroids inhibit the pro-inflammatory cytokine responses, allowing viruses to (re-)activate and undergo lytic proliferation unchecked [60]. This link between elevated corticosteroid levels and herpesvirus reactivation has been proven experimentally (horses: [61]; captive reindeer: [62]) and through observation (e.g., humans: [4,63]; captive Grévy’s zebras (*Equus grevyi*): [3]), but has not been investigated in free-living wildlife populations. Herpesvirus reactivation triggered by stress has been widely confirmed naturally and experimentally by corticosteroid injection in humans and domestic animals [3,61]. Linking stress and viral reactivation in wildlife, however, is particularly challenging due to the difficulties of monitoring individual stress levels in the field, and typically this relationship can only be confirmed experimentally by taking subjects into captivity [62]. Using indicators that have been linked to stress hormone levels in previous studies can thus provide an informative way to study the relationship between stress and herpesvirus reactivation in free-ranging wildlife.

Our finding of no sex-related effect on herpesvirus reactivation in adults is at odds with our recent survey of Irish badger populations [22], where the rate (over 82.5%) of genital MusGHV-1 DNA presence in adult males was significantly higher than in females (47.5%), during the peak (postpartum) mating season (i.e., from mid-January to mid-February). This implies not only a link to mating activity, but also a mechanism enhancing sexual transmission where sex-biases in viral shedding in badger genital tracts are most pronounced during the mating season. This corroborates our finding in the same study where males with more spermatozoa have a higher detection rate of genital MusGHV-1 DNA [22]; linking higher sexual activity to higher STI prevalence as predicted by theory [64] and reported also in many other studies [65,66,67,68]. We were unable to test effects relating to badger mating behaviour explicitly because we did not trap individuals during the mating season, which coincides with late pregnancy and neonatal cub care, where we sought to avoid stressing mothers and to avoid depriving cubs of maternal care. Nevertheless, reactivation rate in autumn, when food sources are most abundant, and badgers undergo a period of reproductive quiescence [69], and thus experience less implicit stress, was significantly lower compared only to summer, but not to spring. This suggests that other factor(s) (e.g., sex hormone cycles [58], oxidative stress [70], and/or genital microbiome [71] and linked coinfections [72]) might also be affecting reactivation rates, beyond the scope of our current study.

Our results also show that social group structure can affect the risk of genital herpesvirus shedding, particularly the higher proportion of cubs within a co-resident social group. This may be because badger cubs generally carry higher pathogen burdens than adults (particularly lice *Trichodectes melis* and coccidia *Eimeria melis*) [73,74], and thus increase per capita immunity burden among all badgers resident in the respective sett.

## 5. Conclusions

Our study demonstrates, for the first time in the wild, the link between host stressors and herpesvirus shedding, which likely reflects both primary infection and reactivation from latency according to host age. Amplified stress levels induced by human disturbance as well as food insecurity and more frequent severe weather events arising from human-induced rapid environmental change could therefore not only increase the risk of disease development and promote pathogen transmission within a population, but also negatively impact host reproductive fitness through localised herpesvirus reactivation in the reproductive tract. Careful monitoring of endemic latent virus infection as well as surveillance for possible newly emerging strains should therefore be included when planning in situ and ex situ conservation programmes for endangered species.

## Figures and Tables

**Table 1 biomolecules-11-00716-t001:** Summary of sampling effort (badgers with unknown age were assigned to age group by tooth wear according to the method described by Bright Ross et al. [46]).

		Spring		Summer		Autumn		
Age Group	Age	Female	Male	Female	Male	Female	Male	Total
Cub	0	15	15	10	4	9	9	62
Yearling	1	1	3	2	2		3	11
Young	2	7	7	7	9	5	6	41
	3	4	8	3	5	2	8	30
	4	2	5	4	5	2	8	26
Old	5	2	1	6	1	3	1	14
	6	3	1	3	1	3		11
	7		2		2		2	6
	Unknown	1	1	1	1		2	6
Very old	8	4	4	4	3	1	3	19
	9	2	1	2	1	1	1	8
	10	3	1	3		4		11
	11							0
	12							0
	13	1	1		1	1		4
	Unknown		1				1	2
	Total	45	51	45	35	31	44	251

**Table 2 biomolecules-11-00716-t002:** Overview of MusGHV-1 DNA detection rate in badger genital tract and univariate logistic regression analysis. Formula: MusGHV-1 ~ Variate + (1|Tattoo); number of observations: 251; groups by tattoo number: 150.

Variable	Positive	Total	%	95% CI	Odds Ratio (OR)	OR 95% CI	*p*-Value
Sex							
Male	45	130	34.62%	27–43.1%			
Female	44	121	36.36%	28.3–45.2%	0.93	0.55–1.56	0.774
Season							
Spring	33	96	34.38%	25.6–44.3%	1.36	0.7–2.65	0.368
Summer	35	80	43.75%	33.4–54.7%	2.02	1.02–4.03	0.044
Autumn	21	75	28.00%	19.1–39%			
Age group							
Juvenile (<2 years old)	33	73	45.21%	34.3–56.6%	5.28	1.85–15.08	0.001
Young (2–4 years old)	30	97	30.93%	22.6–40.7%	2.87	1.02–8.08	0.046
Old (5–7 years old)	5	37	13.51%	5.9–27.8%			
Very old (>7 years old)	21	44	47.73%	33.8–62%	5.9	1.92–17.78	0.002
Body condition ^a^							
Body condition score (1–5)		182			0.64	0.45–0.9	0.012
Social group size							
Total		251			1.02	0.94–1.1	0.653
Adult		251			0.96	0.88–1.05	0.405
Cub		251			1.18	1.01–1.38	0.035
Cub percentage per sett							
Low (<30%)	58	191	30.37%	24.3–37.2%			
High (>30%)	31	60	51.67%	39.3–63.8%	2.35	1.03–4.24	0.004
Lactational status ^bc^							
Not Lactated	3	10	30.00%	10.8–60.3%			
Lactated	6	18	33.33%	16.3–56.3%	0.9	0.16–4.92	0.856

^a^: Only adults were included in this analysis; ^b^: only females captured in spring were included in this analysis; ^c^: Fisher exact test.

**Table 3 biomolecules-11-00716-t003:** Final general mixed effect model of multivariable logistic regression analysis for juveniles. Formula: MusGHV-1 ~ Sex + Body condition + (1|Tattoo); number of observations: 72; groups by tattoo number: 48.

Group	Estimate	Standard Error	*z* Value	Adjusted OR	95% CI	*p* Value
(Intercept)	−1.204	0.813	−1.482			
Sex						
Female						
Male	−0.985	0.528	−1.863	0.37	0.13–1.05	0.062
Body Condition						
BCS	0.785	0.306	2.565	2.19	1.2–3.99	0.01
Cub percentage						
Low (<30%)	−0.977	0.539	−1.812	0.38	0.13–1.08	0.07
High (>30%)						

**Table 4 biomolecules-11-00716-t004:** Final general mixed effect model of multivariable logistic regression analysis for adults. Formula: MusGHV-1 ~ Sex + Season + AgeGroup + Cub percentage + Sex * Cub percentage + (1|Tattoo); number of observations: 171; groups by tattoo number: 99.

Group	Estimate	Standard Error	*z* Value	Adjusted OR	95% CI	*p* Value
(Intercept)	0.059	1.075	0.055			
Season						
Spring	−0.289	0.584	−0.495	0.74	0.24–2.35	0.62
Summer	0.684	0.520	1.318	1.98	0.72–5.48	0.188
Autumn						
Age Group						
Young	1.767	0.673	2.627	5.85	1.57–21.87	0.009
Old						
Very old	1.75	0.695	2.522	5.77	1.48–22.55	0.012
Body condition						
BCS	−0.60	0.238	−2.513	0.54	0.34–0.88	0.012
Cub percentage						
Low (<30%)	−1.15	0.544	−2.115	0.31	0.11–0.92	0.034
High (>30%)						

## Data Availability

Data are available from Appendix A.

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
