# Peer review of "Patterns of Genital Tract Mustelid Gammaherpesvirus 1 (Musghv-1) Reactivation Are Linked to Stressors in European Badgers (Meles Meles)"

_biomolecules, 2021, doi:10.3390/biom11050716_

Round 1

Reviewer 1 Report

General comments:

  1. In title should consider removing the word "adult" European badgers leaving only "European badgers"
  2. The abstract lacks the molecular target (on the basis of which the virus was identified in biological material).
  3. How many positive results were obtained after the PCR reaction with the specific primer? How many after amplification with substituted front primer. What does representative sequences was selected and deposite in GenBank (only 2 sequences was representative?)
    The description in the nauscript is unclear.
  4. Section 3.3 is not clearly written (page 6, line 204).
    How many samples were sequenced 5 or more, and how many sequences were deposited in the Gen bank (because only 2 were shown). Please add phylogenetic analysis on the basis obtained sequences. It will further increase the substantive value of this manuscript, because it is extremely interesting from the point of the evolution of  this virus.

Minor Comments

  1. Expand the abbreviation "DNApol gene" (DNA polymerase gene) (page 3, line 126)
  2.  Expand the abbreviation "NA" in Table 1 (page 3)
  3. List the manufacturers of all molecular biology reagents (see Materials and Methods section) (page.1, line 23)
  4. Order the time units (in the temperature-time profile) (psee Materials and Methods section) (page.3).

Author Response

Dear reviewer,

We are extremely grateful for your time and effort on reviewing our manuscript, subject to the minor revisions we have implemented here. We appreciate how these constructive suggestions have helped to improve the quality and clarity of our manuscript. We have revised the manuscript according to the reviewer comments, which we detail in the attached file. The main changes include that we have added the missing odds ratio values in Table 2 and changed the archiving location of our study data from Dryad to Supplementary Materials. Other minor changes include English grammar improvement throughout the manuscript, correction to second author’s affiliation, correction to the reference number 46 on the author’s last name in the reference list. 

Sincerely

Ming-Shan Tsai, on behalf of authors.

Reviewer 2 Report

Comments and Suggestions for Authors

In this manuscript, genital swabs were screened for detection of MusGHV-1 DNA and some sequences were deposited in GenBank. Second, data about each animal was analyzed using logistic regression to identify risk factors associated with MusGHV infection. The manuscript is well written, however, a small English and writing review would be beneficial. 

The information about risk factors associated with MusGHV-1 in badgers is limited. This manuscript demonstrated scientific soundness and particularly relevance. I recommend this manuscript for publication with minor revisions.

Material and Methods section:

Line 140: Only two sequences were deposited in GenBank, but in results section (line 205-209) reported five sequenced samples. Which criteria had the authors chosen to select only two?

Results section:

In Table 2: The variable used as a reference category should be added in column “OR”, for example in the “male” line. The body condition in adults was not classified into 5 levels? In variable of body condition and Social group size which reference category was used?

In Tables 3 and 4: In variable of body condition which category was used as reference?

Conclusions section:

Line 299-308: This paragraph should not has references in the text.

References:

Line 512: Incomplete reference.

Author Response

(The authors gave the same response as above.)
